

# A deep learning algorithm to detect coronavirus (COVID-19) disease using CT images

Mojtaba Mohammadpoor[1], Mehran Sheikhi karizaki[2] and Mina Sheikhi karizaki[3]

[1] Electrical and Computer Department, University of Gonabad, Gonabad, Iran
[2] Department of Electrical and Computer Engineering, University of Birjand, Birjand, Iran
[3] Nursing Department, Islamic Azad University of Sabzevar, Sabzevar, Iran

## ABSTRACT

**Background:** COVID-19 pandemic imposed a lockdown situation to the world these past months. Researchers and scientists around the globe faced serious efforts from its detection to its treatment.

**Methods:** Pathogenic laboratory testing is the gold standard but it is time-consuming. Lung CT-scans and X-rays are other common methods applied by researchers to detect COVID-19 positive cases. In this paper, we propose a deep learning neural network-based model as an alternative fast screening method that can be used for detecting the COVID-19 cases by analyzing CT-scans.

**Results:** Applying the proposed method on a publicly available dataset collected of positive and negative cases showed its ability on distinguishing them by analyzing each individual CT image. The effect of different parameters on the performance of the proposed model was studied and tabulated. By selecting random train and test images, the overall accuracy and ROC-AUC of the proposed model can easily exceed 95% and 90%, respectively, without any image pre-selecting or preprocessing.

## INTRODUCTION

Severe acute respiratory syndrome coronavirus 2 (SARS-CoV-2), simply called corona virus or COVID-19, is currently one the most life-threatening problems around the world. Coronavirus disease 2019 (COVID-19) is a highly infectious disease caused by severe acute respiratory syndrome coronavirus 2 (*Wang et al., 2020a*). The disease first originated in 31 December 2019 from Wuhan, Hubei Province, China and since then it has spread globally across the world. The cumulative incidence of the causative virus (SARS-CoV-2) is rapidly increasing and has affected 196 countries and territories and on 4 May 2020, a total of 3,581,884 confirmed positive cases have been reported leading to 248,558 deaths (Worldometer: Coronavirus, https://www.worldometers.info/coronavirus/). The impact is such that the World Health Organization (WHO) has declared the ongoing pandemic of COVID-19 a public health emergency of international concern (*Daksh, 2020*).

The pandemic caused by COVID-19 has major differences from other related viruses, such as Middle East Respiratory Syndrome (MERS) and Severe Acute Respiratory

Corresponding author
Mojtaba Mohammadpoor,
mohammadpur@gonabad.ac.ir

Syndrome (SARS), which is its ability to spread rapidly through human contact and leave nearly 20% infected subjects as symptom-less carriers (*Mallapaty, 2020*).

Pathogenic laboratory testing is the gold standard but it is time-consuming; therefore, other diagnostic methods are needed to detect the disease in a timely manner. COVID-19 makes some changes in CT images. *Mahmoud et al. (2020)* have analyzed recent reports and stated that the sensitivity of RT-PCR in diagnosing COVID-19 is 71% while sensitivity of CT is 98%. It is possible that small changes in CT images may be neglected during visual inspection, and we hypothesized that an artificial intelligence method might be able to detect COVID-19's positive cases and provide a clinical diagnosis ahead of the pathogenic test, thus saving critical time for disease control.

The main contribution of this paper was to propose a prediction model based on convolutional neural network (CNN) deep learning method, which is able to be trained by some CT images of corona virus infected lungs and CT images of healthy lungs. The trained model is then able to classify any new CT image as positive and negative COVID-19 at a faster speed.

## Related works

Several efforts have performed by researchers in detecting coronavirus affected cases using radio graphical images. Alibaba has developed AI solutions to predict the duration, size and peak of the outbreak, which were tested in real world in various regions of China and claimed to have 98% accuracy (*Huang et al., 2020*).

As the COVID-19 virus affects the lungs of peoples, some deep learning studies have proposed to detected the disease by processing chest X-ray and CT images of lung (*Jaiswal et al., 2019*). A deep learning model for detecting pneumonia was proposed in *Stephen et al. (2019)*. Their suggested model consisted of convolution layers, dense blocks, and flatten layers. Their input image size is 200 × 200 pixels. Their final success rate is 93.73%.

*Chouhan et al. (2020)* proposed a deep learning model for classifying the pneumonia images into three classes, namely: bacterial pneumonia, virus pneumonia, and normal images. In the first step, they proposed some preprocessing methods to remove noise from the images. Then, they applied an augmentation technique on the images before using them for training their model. Their overall classification accuracy is 96.39%.

*Wang et al. (2020b)* used pathogen-confirmed COVID-19 cases (325 images) and 740 images diagnosed with typical viral pneumonia. Their internal validation reached to an overall classification accuracy of 89.5%. Their external testing dataset reached to an overall accuracy of 79.3%.

*Toğaçar, Ergen & Cömert (2020)* have proposed a deep learning method to classify chest X-ray images to detect corona virus infected patients. Their dataset consisted of three classes, namely: normal, pneumonia and coronavirus images. They achieved to 99.27% classification rate.

*Zahangir et al. (2020)* proposed a multi task deep learning algorithm for this purpose. They used and compared CT scan and X-ray images in their model. They achieved around 84.67% testing accuracy from X-ray images and 98.78% accuracy in CT-images,

meaning that CT scan images are more accurate. They have also tried to determine the percentage of infected regions in CT and X-ray images.

*Zheng et al. (2020)* proposed a 3D deep neural network to predict the probability of COVID-19 infectious. They have used 499 CT volumes for training and 131 CT volumes for testing. Their algorithm reached to 90.1% overall accuracy.

*Gifani, Shalbaf & Vafaeezadeh (2020)* proposed an ensemble method that is using majority voting of the best combination of deep transfer learning of some pre-trained convolutional neural networks. They applied their model on a CT dataset comprising of 349 positive and 397 negative cases and reached to 85% accuracy.

*Mukherjee et al. (2020)* proposed a deep neural network architecture for analyzing both CT Scans and chest X-rays. They achieved an overall accuracy of 96.28% by using their own dataset.

Performance of different deep learning methods have compared together by applying them on pneumonia X-ray images in *Baltruschat et al. (2019)*.

# MATERIALS AND METHODS

Artificial intelligence improves the representations needed for pattern recognition using a machine composed of multiple layers, uses raw data as input (*Goodfellow et al., 2016*). Deep learning is a semi-supervised technique for labeling datasets. For instance, if a deep network is fed with several tumor cells, it can interpret an image to detect insignificant aspects (*Li, 2017*).

Since the last few years, deep learning techniques completely changed the scenario of many research fields by promising results with highest accuracy, especially, in medical image processing fields, such as retina image, chest X-ray, and brain MRI images (*Mahmud et al., 2018*; *Harsono, Liawatimena & Cenggoro, 2020*).

## Convolutional neural networks

Among deep learning classifiers convolutional neural networks (CNN) have more usage in computer vision and medical image analysis tasks compare to others, and it is proved that it has better results (*Panwar et al., 2020*). CNN, as other types of artificial neural network models, has multiple layers and it can process data effectively and achieve high accurate results. Convolution, pooling, flattening, and fully connected layers are consisting CNN structure (*Goodfellow et al., 2016*). CNN can extract the features from the images individually, and then classify them. This unique characteristic can applied on medical images and provides a great support in the advancement of health community research (*Choe et al., 2019*).

Convolutional neural networks models have self-learning abilities helps them to achieve superior and human-like classification results on multi-class problems (*Ucar & Korkmaz, 2020*).

Convolutional neural networks models had been used in different applications and achieve amazing results (*Le & Nguyen, 2019*). In general, they compromised of input, feature extraction and output layers. Feature extraction stage can have several repeated convolution layers, rectified linear units and pooling layers. Convolution layers could

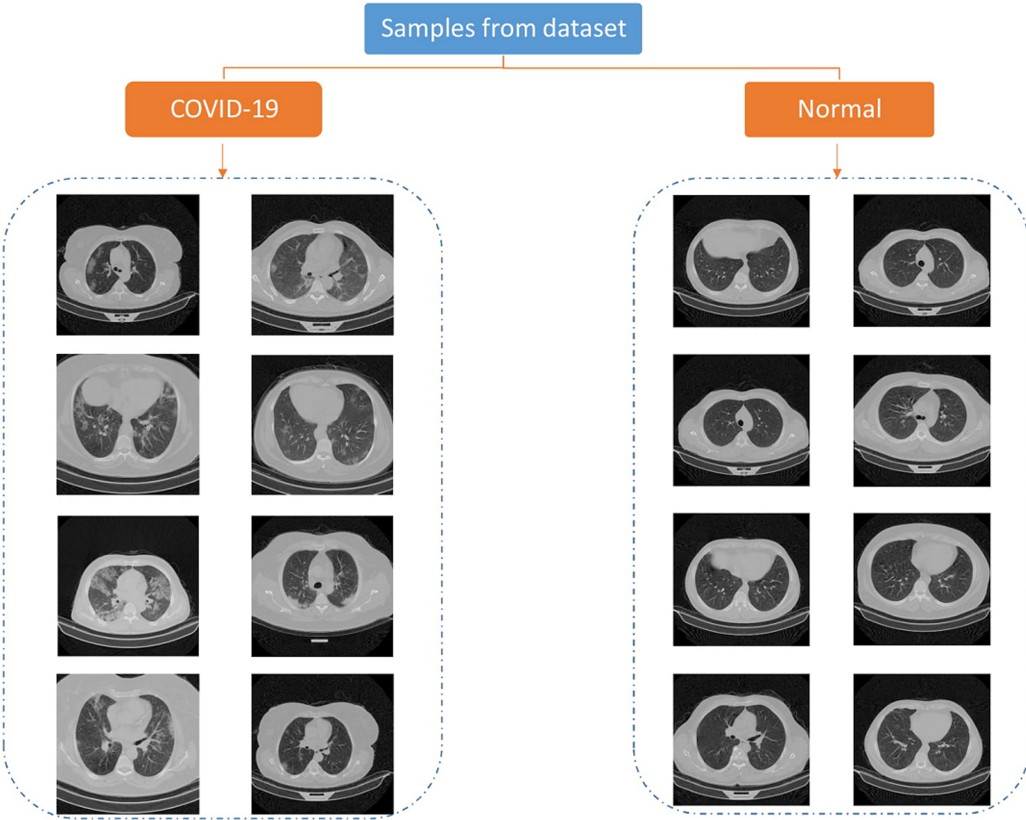

**Figure 1** Samples of COVID-CTset images (*Rahimzadeh, Abolfazl & Seyed, 2020*).

detect different patterns, such as textures, edges, shapes etc. in images (*Jang et al., 2018*, *Raghu et al., 2020*). They also have multilayer perceptrons (fully connected) which all neurons in each layer are connected to all neurons in the next layer. This hierarchical structure provides high-level feature maps and improved overall accuracy (*Ucar & Korkmaz, 2020*).

### Data collection

The data used in this paper is downloaded from publicly available dataset (*Rahimzadeh, Attar & Sakhaei, 2020*). They have collected 15,589 CT images of 95 positive patients and 48,260 images of 282 negative persons. The pictures are 16bit tiff format and 512 × 512 size. Each person has three folder, each folder includes some images representing a breath sequence. Figure 1 is showing some image samples.

In some images of a breath sequence, the inside of the lung is visible. In some of them (e.g., first and lost images of a sequence), inside of the lung is not clear. Figure 2 shows some sequential images.

### Proposed method

A deep learning model based on convolutional neural network (CNN) is proposed in this paper to distinguish positive and negative COVID-19 cases. In some researches some

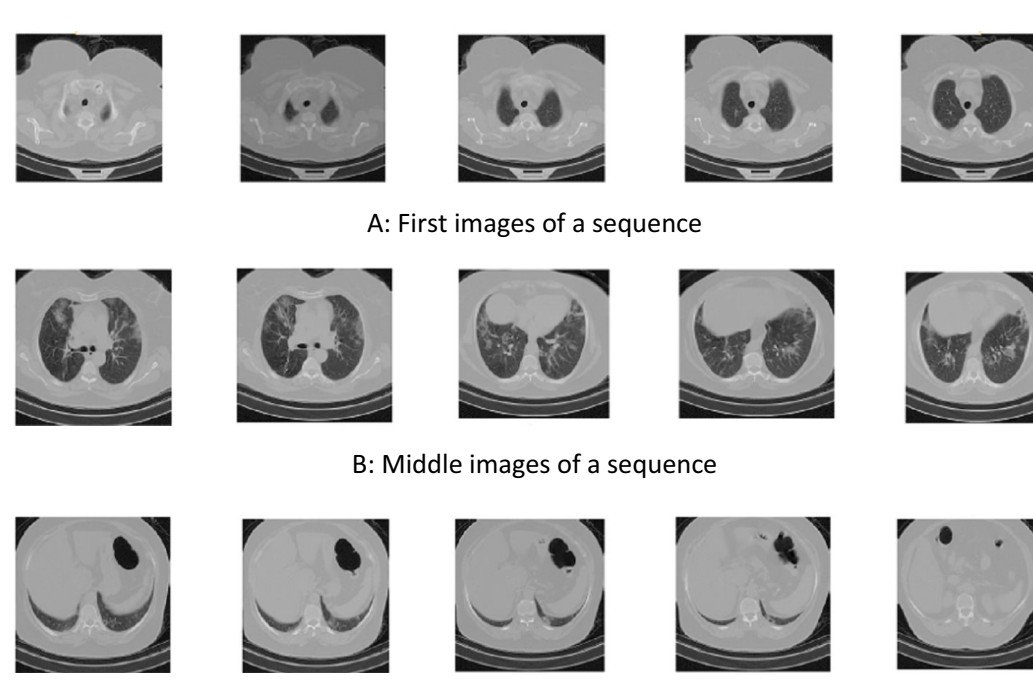

**Figure 2 Samples of sequential COVID-CTset images (*Rahimzadeh, Abolfazl & Seyed, 2020*).**
(A) First images of a sequence, (B) Middle images of a sequence, (B) Last images of a sequence.

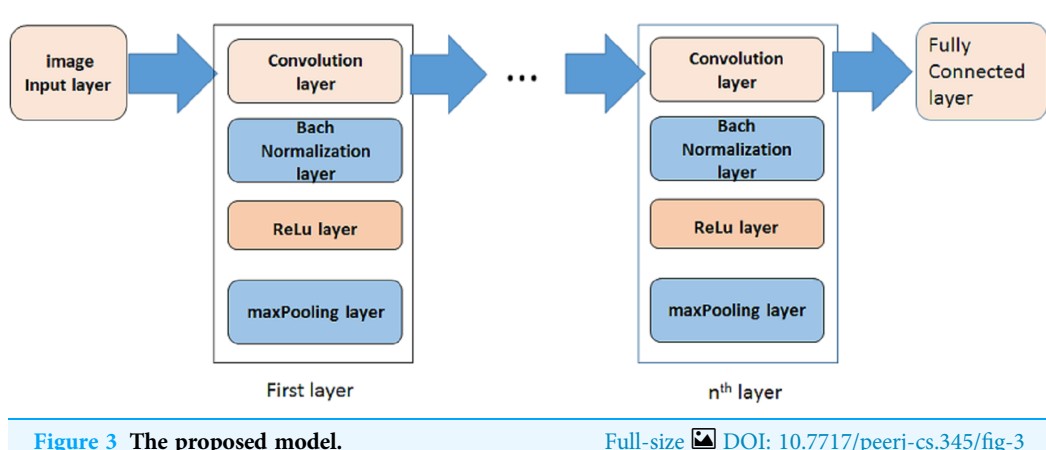

**Figure 3 The proposed model.**

preprocessing stages are applied on images to select special images of a breath sequence or highlight lung infected area, before entering them to the classification algorithm (*Rahimzadeh, Attar & Sakhaei, 2020*). In order to have a fully automated algorithm, in this paper no preprocessing, preselecting or ROI selecting is performed on the images. Figure 3 is showing the proposed model. As it is shown, it is consisted of three steps. In each step a convolution layer (Conv) is used. It is a 2-D convolutional layer which applies sliding convolutional filters to the input image. The layer moves the filters along the input and convolves the input by them vertically and horizontally, and computes the dot product of the input and the weights, and then adds a bias term. In our proposed model, the size of

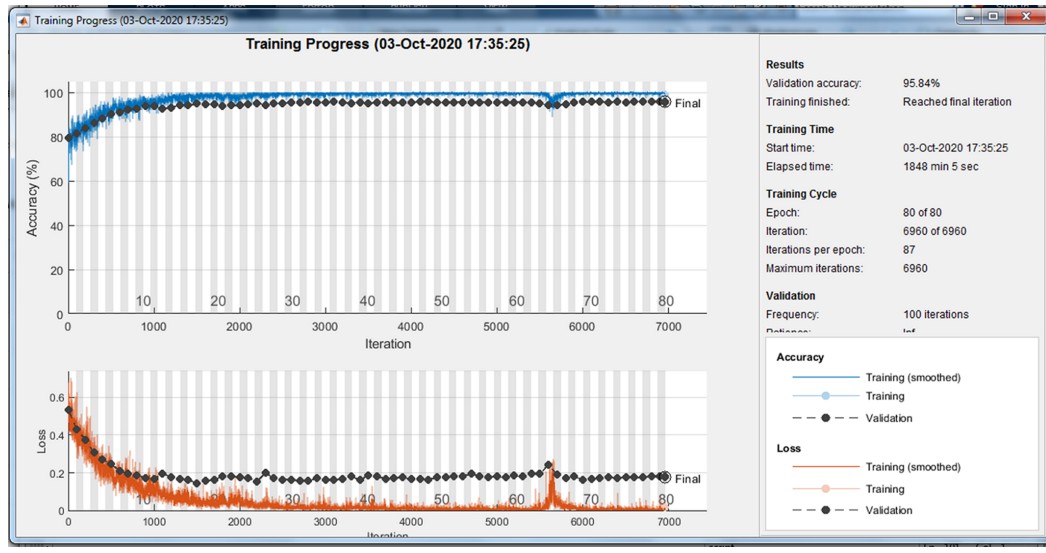

**Figure 4  Training progress of running the model.**  

used filter is selected as 3 × 3. The number of filters are selected as 8, 16, 32, 64 … for other steps.

To reduce sensitivity of CNN to network initialization and speed up its training, a batch normalization layer is used between convolutional layer and nonlinearities. It normalizes each input channel across a mini-batch.

A rectified Linear Unit (ReLU) layer is used in each step to perform a threshold operation to each element of the input, meaning that each value less than zero is set to zero.

A max pooling layer is used in each step to run down-sampling by dividing the input into rectangular pooling regions, and computing the maximum of each region.

In order to evaluate the proposed method, cross-validation technique is performed. For this purpose, the images of each category (i.e., positive or negative) are divided into two groups, namely train, and test. Number of images in each group depends on application. In this paper, the algorithm is performed several times using different percentages. More training images imposed more processing time and leads higher accuracy. A trained network could process any individual image immediately.

## RESULTS

The original images have a size of 512 × 512. In order to reduce the processing border, images with reduced dimensions can be used. Some training options should be defined for training the model. In this paper stochastic gradient descent with momentum (SGDM) optimizer is used. Initial learn rate is selected as 0.001. Maximum number of epochs can affect the training time, as well as accuracy.

Because of randomly selection of train and test images, the model is launched several times. Figures 4–6 are showing results of one running the algorithm. In this sample run, 50% of images in each category are selected randomly for training the model. Others are used for evaluating it. For this purpose, a total number of 2,297 and 8,961 images are selected in positive and negative categories, respectively. The images are resized into 200 ×

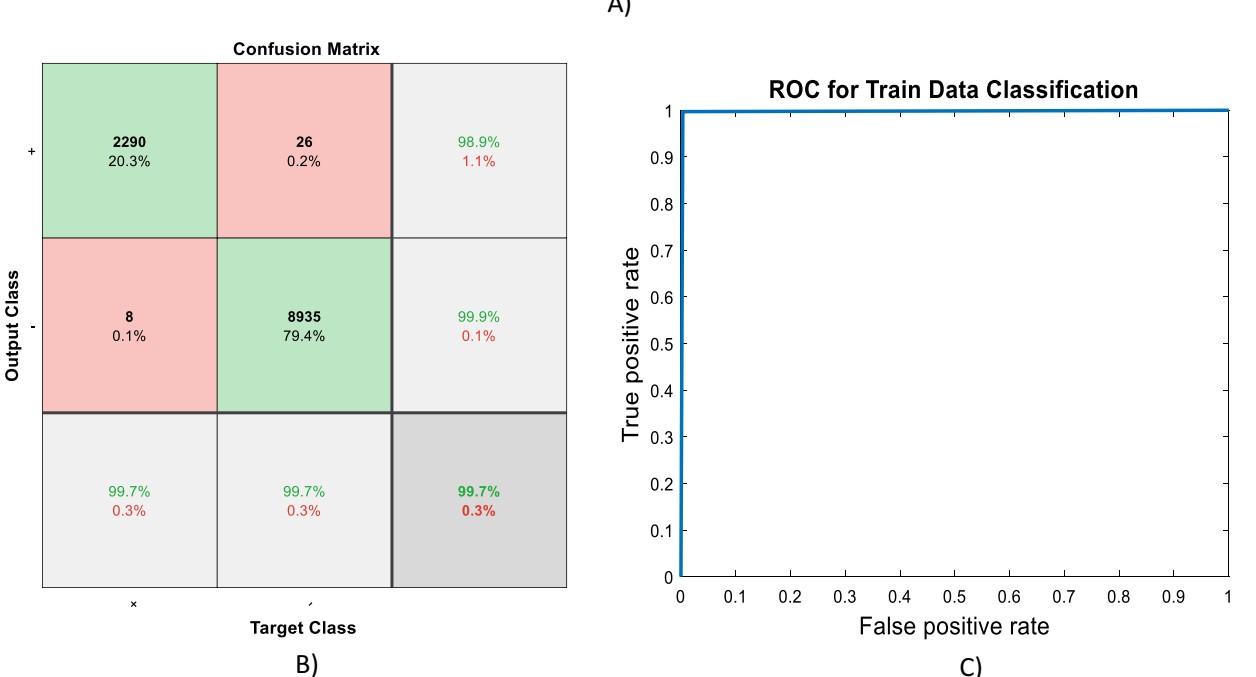

**Figure 5** (A) Confusion matrix definitions (*Wikipedia, 2020*), (B) Confusion matrix and, (C) ROC curve of evaluating training data.

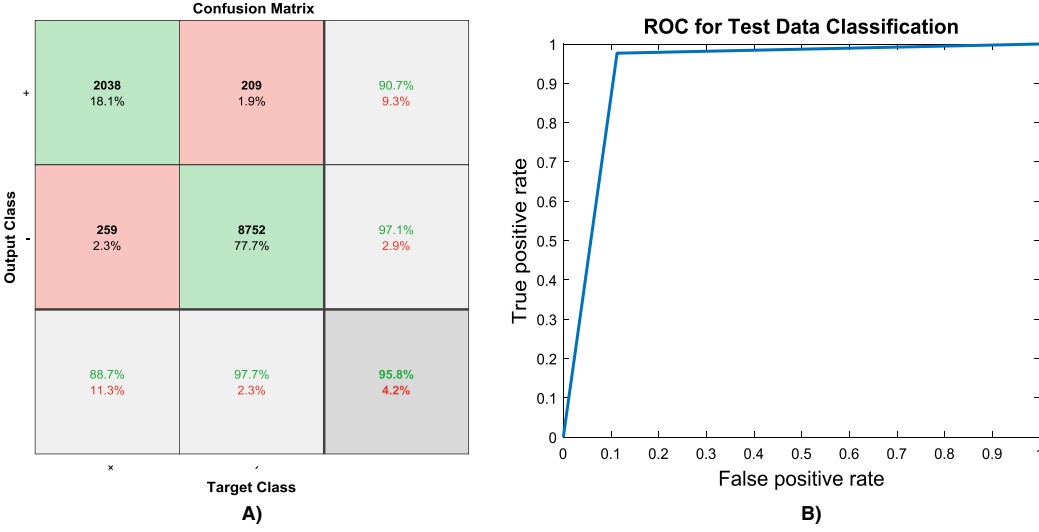

**Figure 6** (A) Confusion matrix and, (B) ROC curve of evaluating test data.

**Table 1** Summarized results of some runs of the proposed algorithm.

| Experiment # | Model adjustments | | | | Results | | | | | |
|---|---|---|---|---|---|---|---|---|---|---|
| | Image size | Training percentage | Max Epochs | No of Conv layers | Training accuracy | Training ROC_AUC | Test accuracy | Test ROC_AUC | All data accuracy | All data ROC_AUC |
| 1 | 200 × 200 | 50% random | 80 | 7 | 0.997 | 0.997 | 0.958 | 0.932 | 0.978 | 0.964 |
| 2 | 512 × 512 | 50% random | 40 | 7 | 0.995 | 0.993 | 0.948 | 0.916 | 0.972 | 0.954 |
| 3 | 512 × 512 | 50% random | 80 | 7 | 0.997 | 0.993 | 0.952 | 0.907 | 0.975 | 0.950 |
| 4 | 75 × 75 | 60% random | 40 | 6 | 0.987 | 0.979 | 0.943 | 0.908 | 0.969 | 0.951 |
| 5 | 50 × 50 | 50% random | 40 | 5 | 0.965 | 0.937 | 0.919 | 0.858 | 0.942 | 0.898 |
| 6* | 50 × 50 | 50% mirror | 40 | 5 | 0.975 | 0.947 | 0.916 | 0.850 | 0.945 | 0.898 |

**Note:**

\* In this experiment, the training and test data of experiment #5 were exchanged.

200 and used for train the deep learning model. Seven convolution layers are used in this case, and maximum epoch is selected as 80. Confusion matrices and Receiver Operating Characteristics (ROC) of the model are shown in Figs. 5 and 6, for evaluating training, test and all data portions, respectively. Total accuracy and Area Under the Curve (AUC) of the ROC curves are shown in the first row of Table 1. Results of running the algorithm by other different parameters are also summarized in Table 1.

## DISCUSSION

Some different parameters can affect the model performance. The first parameter is the image sizes. Large images have more details, hence it is expected to have better results, as shown upper rows in Table 1 have better results compared to lower rows. Second parameter is percentage and method in dividing dataset into train and test category. More training data normally will be caused into higher accuracy. Random selection is selected in all experiments listed in Table 1. In order to show the robustness of the model,

experiment #6 is performed over the opposite data portions of experiment #5, meaning that the model is trained by testing data of experiment #5 and then evaluated by train portion. As it shown the results are reasonable. Another parameter is the number of epochs that deep learning model is performed. More epochs will be caused to better training the model. The last parameter is the number of convolution layers. A deeper network certainly will be well trained. In case of small images, the number of convolution layers may be limited due to padding procedure. Hence, fewer convolution layers are used in $4^{th}$, $5^{th}$, and $6^{th}$ experiments. While using bigger images, this limitation is removed. In the first, second and third experiments in Table 1, seven convolution layers are used. More layers are not test but it is expected that they will have better results. Another parameter is the bit numbers of the images. Originally, the images are 16 bit, in this study they changed to 8 bit.

Eventually, as it shown the overall accuracy and ROC-AUC of the proposed model can easily exceed 95% and 90%, respectively. It should be considered that in this research all CT images during a breath cycle is used, since the inside lung and also infected area can be seen in just few images, the accuracy rate is adequately high which makes it a robust model for detecting CVID19 patients. It is expected that the accuracy increase to 100% by adjusting some parameters, but these parameters can increase the model training time. From an applicable view of point, the model can be trained separately in high performance computers, and then, the trained model be used by doctors, because the trained model can process any individual image in a moment and predict its label almost immediately.

## CONCLUSIONS

Detecting COVID19 positive cases from CT scan images would be helpful for doctors to detect the patients without performing timely and costly molecular tests. In this paper, a machine learning model based on deep learning is proposed for this purpose. The proposed model is evaluated by running it several times on a publicly available CT images dataset. Some percent of images are selected randomly and used for training the proposed model, while the model is evaluated using the remained images. Other adjustable parameters are also discussed. The results implies the ability of the proposed model in classification of images. The overall accuracy and ROC-AUC of the proposed model can easily exceed 95% and 90%, respectively, which makes it a strong CAD tool for use by doctors.

### Funding
The authors received no funding for this work.

### Competing Interests
The authors declare that they have no competing interests.

## Author Contributions

- Mojtaba Mohammadpoor conceived and designed the experiments, performed the experiments, analyzed the data, performed the computation work, prepared figures and/or tables, authored or reviewed drafts of the paper, and approved the final draft.
- Mehran Sheikhi karizaki conceived and designed the experiments, performed the experiments, prepared figures and/or tables, and approved the final draft.
- Mina Sheikhi karizaki conceived and designed the experiments, prepared figures and/or tables, data collection and preparation, and approved the final draft.

## Data Availability

Data are available in Rahimzadeh, Attar, and Sakhaei (2020) and at GitHub: https://github.com/mr7495/COVID-CTset.

Data are also available at Figshare:

Mohammad, Rahimzadeh, (2021): COVID-CTset part1. figshare. DOI 10.6084/m9.figshare.13668596.v1.

Mohammad, Rahimzadeh, (2021): COVID19-CTset part2. figshare. DOI 10.6084/m9.figshare.13669969.v1.

Mohammad, Rahimzadeh, (2021): COVID19-CTset parts 3~5. figshare. Figure. DOI 10.6084/m9.figshare.14174606.v1.

## Supplemental Information

Supplemental information for this article can be found online at http://dx.doi.org/10.7717/peerj-cs.345#supplemental-information.

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
