# Peer review of "A deep learning algorithm to detect coronavirus (COVID-19) disease using CT images"

_PeerJ Computer Science, doi:10.7717/peerj-cs.345_

## Round 0.1 · original submission · Major Revisions

Unfortunately, your article is unsuitable for publication in its present form but we encourage you to enhance and resubmit it following the major revision suggestions.

Reviewer 1 ·

Basic reporting

Authors uses a deep learning approach to classify coronavirus CT images with a being careful of accuracy and computational speed. Language is clear and easy understandable. Literature is well referenced. Figures are clear and describe well the problem, data, methodology and results. Deep learning method is very simple, but it seems to be effective.

Experimental design

It is in the scope of PeerJ. It uses a very simple deep learning model composed of three layers, without preprocess data. Cross validation is used in the right way to support the results.

Validity of the findings

It seems that the model can learn to detect healthy patients and those suffering from COVID-19 with a good accuracy.

I have just a doubt on how it works on data, because they are highly unbalanced, but it seems to not affect the results. I suggest to add a reflection on the validity of the output supported by the good results shown in the cross-validation.

Reviewer 2 ·

Basic reporting

The authors proposed a deep-learning based computer-aided detection (CADe) scheme using a convolutional neural network (CNN) to detect coronavirus disease (COVID-19) using a publicly available CT database.
The topic is very important however I disappointed the construction of study, detailed discussion, and in writing and presentation of results.

Experimental design

# Materials & Methods
## Convolutional Neural Network (CNN)
This paragraph contains only the general of CNN. This is redulant.

# Proposed Method
Figure 3 and Figure 4 are opposite.

# Results
This paragraph is not the result. Generally, computer-aided diagnosis schemes for medical imaging were evaluated using receiver operating characteristic (ROC) analysis. I think the authors need to provide more detailed results.

“Fig.4 is showing a sample result.” – What is “a sample result”?

# Discussions
“As it shown the overall accuracy of the proposed model is more than 84%”- Where is it shown?

Validity of the findings

There are some recent researches to detect COVID-19 using a deep learning algorithm. The authors used a publicly available COVID-19 dataset, so you should compare the proposed model and the state-of-the-art detection algorithm in terms of the detection accuracy. The current presentation remains unconvincing.

Additional comments

While I appreciate the effort of your work presented and the significance of detecting COVID-19, I think the authors needs to improve the focus of the paper and provide more information on the methods used and the results.

Reviewer 3 ·

Basic reporting

In this manuscript authors present a simple CNN model to classify CT images for coronavirus. Language is clear, even though there are some minor typos.
The Convolutional Neural Network used in this paper is really simple and almost shallow but it does the job pretty well.

Experimental design

The approach is well described and there's rigorous investigation proven by the rich bibliography.
Cross-validation is used in the right way and the model performs well even though no pre-processing is used.

Validity of the findings

This paper presents a model that can detect COVID-19 affected patients starting from CT images with a reasonably good amount of accuracy.
A deeper model would have required more data, so the proposed approach is a valid one.

---

## Round 0.2 · Major Revisions

The previous reviewers were invited to re-review this revision but none has responded.

The authors enhanced the manuscript but in this period further researches has been conducted on the this topic. Further information should be introduced:

- references to recent works
- comparisons with other Machine Learning approaches
- use more available data sets

---

## Round 0.3 · accepted · Accept

The article is suitable for publication.